# Prognostic value of plasma pentraxin 3 levels in patients with septic shock admitted to intensive care

S. Perez-San Martin[1], B. Suberviola[2]*, M. T. Garcia-Unzueta[1], B. A. Lavin[1], S. Campos[2], M. Santibañez[3]

1 Department of Clinical Biochemistry, University Hospital Marqués de Valdecilla-IDIVAL, Santander, Spain, 2 Intensive Care Department, University Hospital Marqués de Valdecilla-IDIVAL, Santander, Spain, 3 Health Research Institute Valdecilla-IDIVAL, School of Nursing, University of Cantabria, Santander, Spain

* borja.suberviola@scsalud.es

## Abstract

### Objective

To evaluate the usefulness of a new marker, pentraxin, as a prognostic marker in septic shock patients.

### Materials and methods

Single-centre prospective observational study that included all consecutive patients 18 years or older who were admitted to the intensive care unit (ICU) with septic shock. Serum levels of procalcitonin (PCT), C-reactive protein (CRP) and pentraxin (PTX3) were measured on ICU admission.

### Results

Seventy-five septic shock patients were included in the study. The best predictors of in-hospital mortality were the severity scores: SAPS II (AUC = 0.81), SOFA (AUC = 0.79) and APACHE II (AUC = 0.73). The ROC curve for PTX3 (ng/mL) yielded an AUC of 0.70, higher than the AUC for PCT (0.43) and CRP (0.48), but lower than lactate (0.79). Adding PTX3 to the logistic model increased the predictive capacity in relation to SAPS II, SOFA and APACHE II for in-hospital mortality (AUC 0.814, 0.795, and 0.741, respectively). In crude regression models, significant associations were found between in-hospital mortality and PTX3. This positive association increased after adjusting for age, sex and immunosuppression: adjusted OR T3 for PTX3 = 7.83, 95% CI 1.35–45.49, linear *P trend = 0.024*.

### Conclusion

Our results support the prognostic value of a single determination of plasma PTX3 as a predictor of hospital mortality in septic shock patients.

**Funding:** The authors received no specific funding for this work.

**Competing interests:** The authors have declared that no competing interests exist.

## Introduction

In 2017, the World Health Organization established sepsis as a global priority [1]. The Third International Consensus Definitions for Sepsis and Septic Shock (Sepsis-3) defines sepsis as life-threatening organ dysfunction caused by a dysregulated host response to infection [2]. It is a major cause of admission to the intensive care unit (ICU) and a leading cause of mortality and critical illness worldwide; it is responsible for more than 5 million deaths annually, with an estimated global mortality of 30% [3].

Sepsis and septic shock are medical emergencies, so prompt identification and appropriate management in the initial hours after onset improve outcomes [4], especially among high-risk patients. Thus, biomarkers allowing early stratification and recognition of patients at higher risk of mortality are needed. The two biomarkers that have been most widely studied and used in patients with sepsis are the short pentraxin C-reactive protein (CRP), and procalcitonin (PCT), the prehormone of calcitonin. Nevertheless, even they have limited abilities to distinguish sepsis from other inflammatory conditions or to predict outcome [5, 6]. PCT can be also used to support shortening of the duration of antimicrobial therapy in sepsis patients (weak recommendation, low quality of evidence), according to the latest Surviving Sepsis Campaign (SCC) [4].

In recent years, pentraxin 3 (PTX3), an acute phase protein, has emerged as a promising biomarker of sepsis. It is a prototypical member of the long pentraxin subfamily and a key component of humoral innate immunity. PTX3 is expressed in a number of tissues, particularly dendritic cells and macrophages, in response proinflammatory stimuli [7, 8]. Additionally, it is stored in neutrophil granules and localises in neutrophil extracellular traps [9]. Once released, PTX3 acts by recognising microbes, activating complement and facilitating pathogen recognition by phagocytes, thus promoting pathogen clearance, tuning inflammatory responses and promoting tissue remodelling [7, 8].

In healthy subjects, plasma PTX3 levels are barely detectable (<2 ng/mL) [10], but can quickly rise in inflammatory and infectious conditions [8]. Levels are elevated in critically ill patients, with a gradient from systemic inflammatory response syndrome (SIRS) to sepsis and septic shock [11]. Thus, it has been proposed as prognostic marker for sepsis [11–17]. In a systematic review and meta-analysis, PTX3 significantly predicted disease severity and mortality in sepsis [18].

The aim of this prospective study was to assess the prognostic value of a single determination of PTX3 in septic shock patients in relation to hospital mortality, comparing it with the prognostic value of a single determination of the classical biomarkers CRP and PCT, on ICU admission. We also aimed to evaluate whether its addition to severity scores could improve their prognostic accuracy.

## Materials and methods

### Study design and setting

We performed a single-centre prospective observational study of 75 patients admitted to the medical ICU of Marqués de Valdecilla University Hospital in Spain between April 2015 and April 2016. All consecutive patients 18 years or older who were admitted to ICU with septic shock, according to the Sepsis-2 definition, were eligible to participate. The latest Sepsis-3 definitions were applied and all patients were re-classified according to these new definitions. The criteria for septic shock included the requirement for a vasopressor to maintain a mean blood pressure of 65 mmHg and serum lactate level >2 mmol/L. Exclusion criteria included patients under 18 years of age, as well as individuals with recent cardiac arrest or decision to withdraw

life-sustaining treatment; patients who did not have septic shock at the time of ICU admission but who subsequently developed it were also excluded.

Clinical and demographic characteristics were recorded for all patients, and included age, sex and immunosuppression (AIDS, neutropenia [neutrophil count $<1 \times 10^9$/L], glucocorticoid exposure [$>0.5$ mg/kg for $>30$ d] and/or immunosuppressive or cytotoxic medications, solid organ transplantation, allogeneic or autologous stem cell transplantation, haematological malignancy, or solid tumour). Acute Physiology and Chronic Health Evaluation II (APACHE II) score at 24 h, Simplified Acute Physiology Score (SAPS II) and Sequential Organ Failure Assessment (SOFA) score at admission were also recorded.

The study was approved by the Clinical Research Ethics Committee of Cantabria (CEIC: 2014.159), and written informed consent was obtained from each participating patient or their legal representatives, according to the Declaration of Helsinki.

## Assay methods

An additional blood sample was drawn coinciding with the first extraction for clinical purposes upon admission of the patient to the ICU. Serum and plasma samples were collected as specified for the different assays, and aliquots for this specific study were stored at $-80˚C$ until analysis.

Plasma PTX3 levels were measured by enzyme-linked immunosorbent assay (ELISA) using a commercial kit (Quantikine Human Pentraxin 3/TSG-14; R&D Systems Europe, Abingdon, UK). The sensitivity of the assay is 0.026 ng/mL. Specificity is $<0.5\%$ cross-reactivity observed with available related molecules; $<50\%$ cross-reactivity observed with species tested. Intra-assay reproducibility of the method is $<2.6\%$ and interassay reproducibility is $<6.9\%$.

Serum PCT levels were measured by electrochemiluminescence immunoassay (ECLIA) on a Cobas e411 autoanalyzer (Roche Diagnostics, Mannheim, Germany); serum CRP was determined by immunoturbidimetric assay; and plasma lactate by an enzymatic assay using a Dimension EXL autoanalyzer (Siemens Health Care Diagnostics, Gwynedd, UK).

## Statistical analysis

Categorical and discrete variables were expressed as counts (percentage). Continuous variables were expressed as median and interquartile ranges (IQR). Statistical differences between groups were assessed with the Chi-square test using Yates´ correction or Fisher´s exact test, when appropriate, for categorical variables. Continuous variables were compared with the Mann-Whitney U test. The correlation between continuous variables was calculated using Spearman´s rank correlation coefficient (rho).

To determine and compare the predictive value of the biomarkers and severity scores, receiver-operating characteristic (ROC) curves were constructed and the area under the curve (AUC) was calculated. The outcome variable was in-hospital mortality.

To estimate the strength of associations, the biomarkers and severity scores were divided into dichotomous variables (low versus high values) according to the median and adjusted odds ratios (OR) with their 95% confidence intervals (CI); 95% CIs for in-hospital mortality were calculated using unconditional logistic regression. The following potential confounders were pre-established for inclusion in the models: age (as a continuous variable), sex and immunosuppression status (yes/no).

In addition, exposure-response trends (biological gradient) were estimated, using a logistic regression model with all potential confounders, categorizing the prognostic factors as ordinal variables according to tertiles.

The level of statistical significance was set at 0.05 and all tests were two-tailed. Data were analysed using SPSS statistical software package 24.0 (SPSS, Inc., Chicago, IL, USA).

## Results

### Study population

Seventy-five septic shock patients were included in the study (53 men and 22 women) with a median age at admission of 64 years (Table 1). At ICU admission, median APACHE II, SOFA and SAPS II scores were 22, 10 and 42, respectively. The most common sites of infection were the lungs (41.3%) and abdomen (32%). The ICU mortality rate was 24% and the in-hospital mortality rate was 28%. Non-survivors were more severely ill, as reflected by significantly higher severity scores and higher number of organ dysfunctions on ICU admission;

**Table 1. Baseline characteristics of the study population, in relation to in-hospital mortality.**

| Characteristics | Overall population | Survivors | Non-survivors | p value |
|---|---|---|---|---|
| | n = 75 | n = 54 | n = 21 | |
| Age (years), MD (IQR) | 64 (49–74) | 63 (47.2–70) | 71 (59.5–76.5) | 0.08 |
| Male sex, n (%) | 53 (70.7) | 37 (68.5) | 16 (76.2) | 0.58 |
| Immunosuppression, n (%) | 25 (33.3) | 14 (25.9) | 11 (52.4) | 0.05 |
| Charlson score, MD (IQR) | 4 (2–6.6) | 3.9 (1.1–6.4) | 5.6 (3.4–7.2) | 0.28 |
| Site of infection, n (%): | | | | 0.05 |
| Lung | 31 (41.3) | 22 (40.7) | 9 (42.9) | |
| Abdomen | 24 (32) | 15 (27.8) | 9 (42.9) | |
| Urinary tract | 9 (12) | 9 (16.7) | 0 (0) | |
| Skin-soft tissues | 5 (6.7) | 5 (9.3) | 1 (4.8) | |
| Catheter | 3 (4) | 2 (3.7) | 1 (4.8) | |
| Others | 3 (4) | 1 (1.9) | 1 (4.8) | |
| Nosocomial infection, n (%) | 46 (61.3) | 33 (61.1) | 12 (57.1) | 0,79 |
| APACHE II, MD (IQR) | 22 (15–28) | 19.5 (14–25.2) | 25 (22–30) | 0.006 |
| SAPS II, MD (IQR) | 42 (32–63) | 36 (29.5–50.2) | 61 (53.5–69) | <0.001 |
| SOFA, MD (IQR) | 10 (7–13) | 9 (7–12) | 14 (10–17.5) | <0.001 |
| Organ dysfunction (number), MD (IQR) | 3 (2–4) | 2 (1–3.2) | 4 (3–5) | <0.001 |
| Mechanical ventilation, n (%) | 39 (52) | 19 (35.2) | 20 (95.2) | <0.001 |
| ARDS, n (%) | 11 (14.7) | 3 (5.6) | 8 (38.1) | <0.001 |
| CVVHDF, n (%) | 11 (14.7) | 4 (7.4) | 7 (33.3) | 0.009 |
| Bacteraemia, n (%) | 23 (30.7) | 15 (27.8) | 8 (38.1) | 0.41 |
| SvcO2 (%) | 72 (61.4–78.6) | 72.7 (62.1–78.7) | 74.2 (60–78.6) | 0.99 |
| Lactate (mmol/L), MD (IQR) | 2.7 (1.3–4.6) | 2.1 (1–3.5) | 4.6 (2.7–8.5) | <0.001 |
| PCT (ng/ml), MD (IQR) | 11.8 (1.5–40.6) | 9.6 (1.3–51.3) | 12.9 (3.7–24.7) | 0.47 |
| CRP (mg/dl), MD (IQR) | 22.8 (11.1–25) | 23 (11.1–25) | 20.8 (8.4–25) | 0.56 |
| Pentraxin-3 (ng/ml), MD (IQR) | 63.8 (24.2–187) | 41.8 (14.3–124.3) | 114.4 (56.1–250) | 0.01 |
| ICU mortality, n (%) | 18 (24) | - - - - | - - - - | - - - - |
| Hospital mortality, n (%) | 21 (28) | - - - - | - - - - | - - - - |
| ICU stay (days), MD (IQR) | 4.9 (2.9–9.9) | 5.9 (2.9–9.9) | 4.9 (1.9–9.9) | 0.10 |

Apache II: Acute Physiology and Chronic Health disease Classification System II; SAPS: Simplified Acute Physiology Score; SOFA: Sequential Organ Failure Assessment; ARDS: Acute respiratory distress syndrome; CVVHDF: Continuous venovenous haemodiafiltration; SvcO2: Central venous oxygen saturation; PCT: Procalcitonin; CRP: C-reactive protein; VDBP: Vitamin D binding protein. MD (IQR); median (interquartile range). P results based on Mann–Whitney U test.

mechanical ventilation and haemodiafiltration were required in a higher percentage of non-survivors compared to survivors.

PTX3 levels were higher than normal in all patients, with a median PTX3 level of 63.8 (24.2–187) ng/mL. Median PCT and CRP levels were 11.8 ng/mL (1.5–40.6) and 22.8 (11.1–25) ng/mL, respectively. Plasma PTX3 concentrations were higher in non-survivors compared to survivors (114.4 vs. 41.8 ng/mL, $P$ = 0.01). However, serum PCT and CRP concentrations showed no statistically significant differences between these groups.

At ICU admission, plasma PTX3 levels showed a statistically significant positive correlation with the APACHE II score (rho = 0.402, $P$<0.001), SOFA score (rho = 0.378, $P$ = 0.001), SAPS-S-II score (rho = 0.449, $P$<0.001) and PCT (rho = 0.329, $P$ = 0.005), but not with CRP (rho = 0.157, $P$ = 0.187), ICU length of stay (rho = 0.104, $P$ = 0.82) or hospital length of stay (rho = 0.041, $P$ = 0.73).

## Prognostic value of PTX3: ROC and regression analysis

Among the prognostic factors studied, the best rates of prediction of in-hospital mortality were the severity scores: SAPS II (AUC = 0.81), SOFA (AUC = 0.79) and APACHE II (AUC = 0.73). The ROC curve for PTX3 (ng/mL) yielded an AUC of 0.70, higher than the AUC for PCT (0.43) and CRP (0.48) but lower than the AUC for lactate (0.79) (Table 2) (S1 Fig). Adding PTX3 to the logistic model slightly increased the predictive capacity in relation to SAPS II, SOFA and APACHE II for in-hospital mortality (AUC 0.814, 0.795, and 0.741, respectively) (Table 3).

Tables 4 and 5 show the strength of associations (OR) and exposure-trends (biological gradient) between severity scores and biomarkers, including PTX3, in relation to in-hospital mortality. When categorising the prognostic factors according to the median and tertiles, the results were consistent with the predictive capacity determined using the AUCs. With respect to biomarkers, lactate was statistically significantly associated with in-hospital mortality, but the associations were higher for PTX3 compared to PCT and CRP.

Among the severity scores (SOFA, SAPS II and APACHE II), significant crude positive associations were obtained for in-hospital mortality, crude OR at the highest tertile (OR T3): OR T3 for SAPS II = 30.73, 95% CI 3.57–264,49 $P$ trend <0.001; OR T3 for SOFA = 59.80, 95% CI 0.65–19.84, $P$ trend = 0.001; and OR T3 for APACHE II = 5.90, 95% CI 1.37–25.36, $P$ trend = 0.016). After adjusting for age, sex and immunosuppression, these positive associations increased for all severity scores.

Regarding biomarkers, non-significant associations with no dose-response $P$ trends were found for PCT and CRP. In crude regression models, significant associations with significant

**Table 2. Area under the curve (AUCs) with 95% confidence intervals (CIs) for severity scores and biomarkers in relation to in-hospital mortality.**

|  | AUC[a] | (95% | CI) |
|---|---|---|---|
| APACHE II | 0.728 | 0.612 | 0.843 |
| SAPS II | 0.814 | 0.713 | 0.915 |
| SOFA | 0.785 | 0.672 | 0.899 |
| PCT (ng/ml) | 0.534 | 0.398 | 0.671 |
| CRP (mg/dl) | 0.481 | 0.339 | 0.623 |
| Lactate (mmol/L) | 0.794 | 0.692 | 0.895 |
| Pentraxin-3 (ng/ml) | 0.698 | 0.575 | 0.822 |

[a] AUC denotes Area Under the Curve ROC; 95% CI Confidence interval at 95%.

**Table 3. Area under the curve (AUCs) with 95% confidence intervals (CIs) estimated through regression models for different combinations of APACHE II, SAPS II, SOFA and pentraxin-3 in relation to in-hospital mortality.**

|  | AUC[a] | (95% | CI) |
|---|---|---|---|
| APACHE II + Pentraxin-3 (ng/ml) | 0.741 | 0.624 | 0.858 |
| SAPS II + Pentraxin-3 (ng/ml) | 0.817 | 0.719 | 0.916 |
| SOFA + Pentraxin-3 (ng/ml) | 0.795 | 0.689 | 0.901 |

[a] AUC denotes Area Under the Curve ROC; 95% CI Confidence interval at 95%.

dose-response trends were found between in-hospital mortality and PTX3. This positive association increased after adjusting for age, sex and immunosuppression: adjusted OR at the highest tertile (ORa T3) for PTX3 = 7.83, 95% CI 1.35–45.49, $P$ trend = 0.024. Lactate was the biomarker with highest associations: adjusted OR at the highest tertile (ORa T3) for lactate = 36.47, 95% CI 3.85–345.38, $P$ trend <0.001 (Table 5).

## Discussion

In our study, we evaluated the prognostic value of PTX3, PCT and CRP in patients with septic shock defined according to Sepsis-3 criteria. Our findings suggest that plasma PTX3 could be a

**Table 4. Associations between severity scores and in-hospital mortality.**

|  |  | Survivors | Non-survivors |  |  |  |  |  |  |
|---|---|---|---|---|---|---|---|---|---|
| Severity scores | Cut-off points | N = 114 | N = 25 | OR | (95% | CI) | ORa | (95% | CI) |
| *APACHE II (Median)* |  |  |  |  |  |  |  |  |  |
| Low (reference) | ≤ 22 | 38 | 6 | 1.00 | - - |  | 1.00 | - - |  |
| High | 23+ | 16 | 15 | 5.94 | 1.95 | 18.06 | 6.59 | 1.82 | 23.93 |
| *APACHE II (Tertiles)* |  |  |  |  |  |  |  |  |  |
| Low (reference) | ≤ 17 | 23 | 3 | 1.00 | - - |  | 1.00 | - - |  |
| Medium | 18–25 | 18 | 8 | 3.41 | 0.79 | 14.72 | 2.92 | 0.57 | 14.94 |
| High | 26+ | 13 | 10 | 5.90 | 1.37 | 25.36 | 6.22 | 1.18 | 32.67 |
| p linear trend |  |  |  | p = 0.016 |  |  | p = 0.029 |  |  |
| *SAPS II* |  |  |  |  |  |  |  |  |  |
| High (reference) | ≤ 42 | 36 | 2 | 1 | - - |  | 1 | - - |  |
| Low | 43+ | 18 | 19 | 19 | 3.98 | 90.69 | 23.362 | 3.972 | 137.411 |
| *SAPS II (Tertiles)* |  |  |  |  |  |  |  |  |  |
| High (reference) | ≤ 35 | 26 | 1 | 1 | - - |  | 1 | - - |  |
| Medium | 36–56 | 17 | 7 | 10.71 | 1.21 | 94.96 | 11.01 | 1.13 | 107.37 |
| Low | 57+ | 11 | 13 | 30.73 | 3.57 | 264.49 | 48.67 | 4.46 | 530.95 |
| p linear trend |  |  |  | p<0.001 |  |  | p = 0.001 |  |  |
| *SOFA (Median)* |  |  |  |  |  |  |  |  |  |
| High (reference) | ≤ 10 | 36 | 6 | 1 | - - |  | 1 | - - |  |
| Low | 11+ | 18 | 15 | 5.00 | 1.66 | 15.07 | 13.10 | 2.58 | 66.47 |
| *SOFA (Tertiles)* |  |  |  |  |  |  |  |  |  |
| High (reference) | ≤ 8 | 24 | 2 | 1 | - - |  | 1 | - - |  |
| Medium | 9–12 | 20 | 6 | 3.60 | 0.65 | 19.84 | 2.75 | 0.41 | 18.40 |
| Low | 13+ | 10 | 13 | 15.60 | 2.96 | 82.17 | 59.80 | 5.13 | 696.73 |
| p linear trend |  |  |  | p = 0.001 |  |  | p = 0.001 |  |  |

ORa: Odds ratio adjusted for age, sex and immunosuppression.

**Table 5. Associations between biomarkers and in-hospital mortality.**

| Biomarkers | Cut-off points | Survivors N = 114 | Non-survivors N = 25 | OR | (95% | CI) | ORa | (95% | CI) |
|---|---|---|---|---|---|---|---|---|---|
| **PCT (ng/ml) (Median)** | | | | | | | | | |
| Low (reference) | ≤ 11.80 | 30 | 8 | 1.00 | - - | | 1.00 | - - | |
| High | 11.81+ | 24 | 13 | 2.03 | 0.72 | 5.7 | 2.87 | 0.86 | 9.58 |
| **PCT (ng/ml) (Tertiles)** | | | | | | | | | |
| Low (reference) | ≤ 4.00 | 20 | 5 | 1.00 | - - | | 1.00 | - - | |
| Medium | 4.01–22.60 | 15 | 10 | 2.67 | 0.75 | 9.45 | 2.31 | 0.6 | 8.9 |
| High | 22.61+ | 19 | 6 | 1.26 | 0.33 | 4.84 | 1.73 | 0.39 | 7.75 |
| *p linear trend* | | | | *p = 0.753* | | | *p = 0.422* | | |
| **CRP (mg/dl) (Median)** | | | | | | | | | |
| Low (reference) | ≤ 22.70 | 26 | 12 | 1.00 | - - | | 1.00 | - - | |
| High | 22.71+ | 28 | 9 | 0.7 | 0.25 | 1.92 | 0.82 | 0.27 | 2.46 |
| **CRP (mg/dl) (Tertiles)** | | | | | | | | | |
| Low (reference) | ≤ 13.00 | 17 | 8 | 1.00 | - - | | 1.00 | - - | |
| Medium | 13.01–25.00 | 32 | 12 | 0.797 | 0.273 | 2.325 | 0.998 | 0.311 | 3.202 |
| High | 25.01+ | 5 | 1 | 0.425 | 0.042 | 4.263 | 0.39 | 0.033 | 4.65 |
| *p linear trend* | | | | *p = 0.466* | | | *p = 0.615* | | |
| **Lactate (mmol/L) (Median)** | | | | | | | | | |
| Low (reference) | ≤ 24.00 | 33 | 6 | 1.00 | - - | | 1.00 | - - | |
| High | 24.01+ | 21 | 15 | 3.93 | 1.32 | 11.73 | 4.48 | 1.36 | 14.75 |
| **Lactate (mmol/L) (Tertiles)** | | | | | | | | | |
| Low (reference) | ≤ 18 | 26 | 1 | 1.00 | - - | | 1.00 | - - | |
| Medium | 18.01–32.00 | 16 | 7 | 11.38 | 1.28 | 101.22 | 9.53 | 1.00 | 90.40 |
| High | 32.01+ | 12 | 13 | 28.17 | 3.29 | 240.81 | 36.47 | 3.85 | 345.38 |
| *p linear trend* | | | | p<0.001 | | | p<0.001 | | |
| **Pentraxin-3 (ng/ml) (Median)** | | | | | | | | | |
| Low (reference) | ≤ 63.80 | 30 | 8 | 1.00 | - - | | 1.00 | - - | |
| High | 63.81+ | 24 | 13 | 2.03 | 0.72 | 5.7 | 2 | 0.64 | 6.24 |
| **Pentraxin-3 (ng/ml) (Tertiles)** | | | | | | | | | |
| Low (reference) | ≤ 31.90 | 23 | 2 | 1.00 | - - | | 1.00 | - - | |
| Medium | 31.91–116.60 | 16 | 9 | 6.47 | 1.23 | 34.01 | 6.05 | 1.05 | 34.93 |
| High | 116.61+ | 15 | 10 | 7.67 | 1.47 | 39.99 | 7.83 | 1.35 | 45.49 |
| *p linear trend* | | | | *p = 0.015* | | | *p = 0.024* | | |

ORa: Odds ratio adjusted for age, sex and immunosuppression.

potential predictor of mortality, having, according to our results, a superior prognostic value compared to PCT and CRP. Our results also support severity scores—classical markers used to predict mortality—as predictors of mortality, particularly SAPS II. This reproducible accuracy of the severity scores supports the validity of our results for PTX3. We also found that PTX3 was elevated and showed a stronger correlation with disease severity, organ dysfunction and other clinical parameters than CRP or PCT in septic shock.

In accordance with a previous study, CRP remained equally high in both survivors and non-survivors [12]. Although PTX3 and CRP belong to the same pentraxin family, PTX3 differs from CRP in terms of gene organization and localisation, ligand recognition, cellular source and inducing signal [7, 8]. PTX3 is an acute phase protein secreted by various cells in response to proinflammatory signals, unlike the short pentraxin CRP, which is produced in

the liver and induced by IL-6. Under normal physiological conditions, plasma PTX3 levels are low (<2 ng/mL) but increase rapidly in sepsis as a result of neutrophil degranulation—up to 100 ng/mL depending on the severity of disease [19]—with levels maintained through *de novo* production by endothelial cells and some monocytic cells [20, 21]. PTX3 increases within 6–8 h of response to infection, compared to 24–30 h for CRP.

The association between PTX3 and the increase in in-hospital mortality remained after adjusting for the main confounding factors, namely age, sex, and immunosuppression, supporting the independence of this association. Moreover, an exposure-response pattern (a dose-response trend) was found, where higher levels were associated with a higher risk of mortality. Patients with high levels (third tertile) had an approximately eight-fold risk of in-hospital mortality compared to patients with lower PTX3 levels. However, the predictive accuracy of a single PTX3 determination, in relation to the AUC determined by the ROC curves, was lower than the severity scores (SAPS II, SOFA, APACHE II), but superior to the classically used biomarkers (CRP, PCT) in predicting the risk of mortality in septic shock patients. The predictive accuracy in relation to severity scores was slightly increased by adding PTX3 to the logistic models. The clinical relevance of this slight increase merits further attention in future studies, to elucidate whether systemic PTX3 levels would have prognostic value and could help to determine the prognosis of patients with septic shock, complementing disease severity classification systems and other biological markers.

Sepsis occurs when the release of proinflammatory mediators in response to an infection extends beyond the boundaries of the local environment, leading to a more generalised response, so it is difficult to describe it with a single measure. The predictive scoring systems used to predict mortality in ICU patients have major limitations, including poor generalisability, deterioration over time and possibly lead-time bias [22–24], and they also tend to be used more in research than in routine clinical practice. For this reason, although PTX3 did not perform as well as severity scores, especially SAPS II, it should not be excluded for use as a predictor of outcome. CRP is sensitive but not very specific, since it is increased in all inflammatory disorders, while PCT differentiates between infectious and non-infectious causes of critical illness better than CRP [25]. Nevertheless, they provide limited information in relation to patient prognosis in critically ill patients [26, 27]. A meta-analysis of 21 studies with a total of 6007 patients concluded that the initial PCT level was of limited prognostic value in patients with sepsis [28]. In another meta-analysis of 25 studies with 2353 patients, Arora et al. found that, in a subgroup of patients with severe sepsis and septic shock, there was no difference in PCT values between survivors and non-survivors on day 1 ($P = 0.062$) [29]. In addition, serum levels may be altered by clinical and demographic conditions [30–32]. Our data showed a significant correlation between PCT and PTX3 at ICU admission, but PCT was not an independent marker for in-hospital mortality, nor was CRP. At ICU admission, the AUC value of a single PTX3 determination was significantly much better than CRP and PCT in predicting in-hospital mortality. Moreover, PTX3 levels correlated better with disease severity and organ dysfunction than PCT or CRP.

Our study findings are consistent with previous studies that examined PTX3 as a prognostic marker for sepsis [11–17]. Levels of PTX3 have been correlated with disease severity, organ dysfunction and markers of coagulation activation and, when compared with other biomarkers (e.g. IL-6, TNFα and CRP), have shown a stronger correlation with clinical parameters [33]. In a systematic review and meta-analysis, PTX3 was identified as a marker of sepsis severity and predictor of mortality, but with limited specificity [18]. In a prospective study with 112 patients with septic shock, baseline PTX3 levels were an independent predictor of 28-day mortality, unlike CRP and PCT [17]. In contrast to our study, Mauri et al. reported that, although high PTX3 levels over the first 5 days from onset of sepsis were correlated with poorer

outcomes, the initial PTX3 levels at ICU admission did not differ between survivors and non-survivors [12]. Additionally, Caironi et al., in a large multicentre randomised controlled trial, found that plasma PTX3 concentrations were higher in non-survivors compared to survivors on day 1 and were correlated with severity. While PTX3 levels on day 7 showed a significant predictive value for 90-day mortality, PTX3 levels on day 1, after adjustment for all confounders, were not associated with 90-day mortality in patients with severe sepsis or septic shock [15]. These conflicting outcomes may due to the heterogeneity of different study subjects. Both studies were designed with patients with severe sepsis and septic shock according to the first criterion of sepsis based on the concept of SIRS. In addition, patients were randomised at enrolment. In contrast, our clinical study was conducted on a homogeneous group of patients who met the updated definition of septic shock. Additionally, the timing of mortality analysis was different (90-day mortality vs. in-hospital mortality), and in both studies, a sandwich ELISA developed in-house was used to determine PTX3 levels.

It should be pointed out that our results are also consistent with recent prospective studies, according to the Sepsis-3 definitions [34–37]. In contrast to our study, Hu et al. reported that PCT is a moderate predictor of 28-day mortality in patients with sepsis and septic shock.

Our results also showed that elevated lactate levels were highly associated with in-hospital mortality in septic shock, in line with other studies [4, 38–41]. Lactate has been widely used as a marker of altered tissue perfusion. Adult patients with septic shock can be identified using the clinical criteria of hypotension requiring the use of vasopressors to maintain mean blood pressure of 65 mmHg or greater and having a serum lactate level greater than 2 mmol/L persisting after adequate fluid resuscitation [2]. However, an elevated lactate level is a sensitive marker for cellular dysfunction in sepsis [42, 43], but is non-specific, since it can be elevated in other types of shock, such as cardiogenic, obstructive or hypovolaemic shock, and can be affected by several factors, such as liver disease [41, 44].

The present study has several advantages. We performed a prospective study with a very homogeneous group of patients who met septic shock criteria in accordance with the latest Sepsis-3 definitions at the time of admission to the ICU. Furthermore, the prognosis was established very promptly because it was based on the fact that blood samples were obtained on ICU admission.

This study also has some limitations, namely that the generalisability of our findings is constrained by the fact that this was a single centre observational study with a small sample size.

## Conclusions

In summary, our results support the available published studies suggesting the prognostic value of a single determination of plasma PTX3 as a predictor of hospital mortality in septic shock patients, defined according to the latest Sepsis-3 criteria.

Further multicentre studies with larger sample size are needed in order to generalise these results, elucidating the existence of subgroups of patients with specific characteristics in whom this biomarker could demonstrate higher accuracy in assessing the risk of mortality or increased severity. This would allow us to better understand the predictive potential of PTX3 in comparison to other existing biomarkers such as lactate, PCT or CRP or severity scores.

## Supporting information

**S1 Fig. ROC curves for severity scores and biomarkers with respect to 'in-hospital mortality'.**
(DOC)

## Author Contributions

**Conceptualization:** B. Suberviola, M. Santibañez.

**Data curation:** S. Perez-San Martin, B. Suberviola, M. T. Garcia-Unzueta, M. Santibañez.

**Formal analysis:** M. Santibañez.

**Methodology:** B. Suberviola.

**Writing – original draft:** S. Perez-San Martin, B. Suberviola, M. T. Garcia-Unzueta, B. A. Lavin, S. Campos, M. Santibañez.

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
