## [Decision Letter · Decision Letter 0]

9 Oct 2020

PONE-D-20-28810

PROGNOSTIC VALUE OF PLASMA PENTRAXIN 3 LEVELS IN PATIENTS WITH SEPTIC SHOCK ADMITTED TO INTENSIVE CARE

PLOS ONE

Dear Dr. Suberviola,

Thank you for submitting your manuscript to PLOS ONE. After careful consideration, we feel that it has merit but does not fully meet PLOS ONE’s publication criteria as it currently stands. Therefore, we invite you to submit a revised version of the manuscript that addresses the points raised during the review process.

We look forward to receiving your revised manuscript.

Kind regards,

Aleksandar R. Zivkovic

Academic Editor

PLOS ONE

Journal Requirements:

Reviewers' comments:

Reviewer's Responses to Questions

**Comments to the Author**

1. Is the manuscript technically sound, and do the data support the conclusions?

Reviewer #1: Partly

Reviewer #2: Yes

Reviewer #3: Partly

2. Has the statistical analysis been performed appropriately and rigorously? 

Reviewer #1: N/A

Reviewer #2: Yes

Reviewer #3: N/A

3. Have the authors made all data underlying the findings in their manuscript fully available?

Reviewer #1: Yes

Reviewer #2: Yes

Reviewer #3: Yes

4. Is the manuscript presented in an intelligible fashion and written in standard English?

Reviewer #1: Yes

Reviewer #2: Yes

Reviewer #3: Yes

5. Review Comments to the Author

Reviewer #1: - Describing Pentraxin 3 as a new marker for sepsis is not correct , many studies has discussed its role in sepsis . may be the Gap between the study period ( 2015-2016) and the submission for publishing is a factor . this time gap make most of the results of this study NOT Novel which is an important issue for publication.

- A major drawback in this study is lack of control group

-In Table one how can you make a statistical comparison between survival with UTI and non-Survival ( count = zero)

- what were the results of cultures( blood, urine, etc)

- Ptx3 levels increase in other NON-SEPSIS condition as hypoxia, renal insult . DO you exclude those patients??

- what is the initial diagnosis of your patients

- it is recommended to include figures of ROC

Reviewer #2: In the present study, Perez-San Martin et al. present data showing the prognostic potential of PTX3 in a cohort of 75 septic shock patients. They state that adding PTX3 to severity scores increased the predictive capacity of the latter for mortality, and that a single determination of plasma PTX3 on ICU admission was a better prognostic marker compared with PCT and CRP.

The study is relatively limited in terms of number of patients involved, but it confirms the general messages of previous studies on the prognostic potential of PTX3 in sepsis.

Major points:

- The authors state that adding PTX3 to severity scores increased the predictive capacity of the latter for mortality. However, this increase is really limited and AUC reported in Tables 2 and 3 are almost identical. If there is any advantage, is this statistically significant?

-The authors should better underline the novelties of this study compared to papers published in the last years on this marker in sepsis, which are collected in the review by Porte et al (Frontiers Immunol, 2019), in addition to ref 18. Among them, there are studies with larger sample size.

-The authors discuss the difference with results reported in ref 12 and 15, pointing to the value of PTX3 levels at ICU admission found in this study, but not in the studies of ref 12 and 15, where the authors state that levels on day 5 and 7, respectively, correlated with severity, but PTX3 levels on day 1, did not. In this regard, it would be important to discuss criteria for ICU admission, and the time between sepsis diagnosis and PTX3 test, since these factors could explain the difference with previous studies.

-It would be important to add figures, e.g. for AUC and correlation analysis.

-It would be interesting to perform further analysis based on the characteristics of patients. For instance, is there any difference in PTX3 levels/predictive capacity depending on the infection site?

Reviewer #3: The authors discuss the prognostic role of a single determination at the time of ICU admission of 3 biomarkers (PCT, CRP, PTX-3) in septic shock patients. They conclude that, after adjustment for confounding, PTX-3 has a prognostic value superior to the other two: the higher this biomarker, the higher the mortality.

I have several concerns.

1) The authors state that biomarkers are needed to stratify the patients according to the risk of death. Indeed, severity scores gather several clinical characteristic to convert them into a risk of mortality. PCT and CRP, being inflammatory markers, are certainly related to the prognosis, however their clinical use is usually different: early recognition of infection, follow-up of a certain treatment, antibiotic stewardship. Do you really think that such biomarkers could perform better than a severity score in prognostication? In Table 2 and 3 the authors present these findings, but they do not discuss them appropriatedly.

2) The classical biomarker for sepsis prognostication is lactate, which is not discussed at all.

3) Premise: I am not a statistician. The ROC/AUC approach is ok, but I think that a survival analysis should be used rather than a logistic regression. The exposure-response trend with three groups does not add much to the analysis performed with two groups. Table 4 and 5 are redundant.

Additional remarks

- The biological and clinical role of the biomarkers should be clearly presented either in the Introduction or in the Discussion section without repetition

- Try to discuss the discrepancies with Caironi's and Mauri's studies

- Lack of a non-septic control group is not a limitation of the study

- Conclusions, first sentence: PTX-3 is an actual, not potential prognostic factor. The sentence does not mean anything in this way.

6. PLOS authors have the option to publish the peer review history of their article (what does this mean?). If published, this will include your full peer review and any attached files.

Reviewer #1: No

Reviewer #2: No

Reviewer #3: No

---

## [Author Response · Author response to Decision Letter 0]

11 Nov 2020

Reviewer #1: - Describing Pentraxin 3 as a new marker for sepsis is not correct , many studies has discussed its role in sepsis . may be the Gap between the study period ( 2015-2016) and the submission for publishing is a factor . this time gap make most of the results of this study NOT Novel which is an important issue for publication.

- A major drawback in this study is lack of control group

Response: The design is a cohort study: the patients with the lower values (below the median) act as the non-exposed (reference group) in this design, and the outcome (the event) is in-hospital mortality, comparing the risk of this event between exposed and non-exposed.

-In Table one how can you make a statistical comparison between survival with UTI and non-Survival ( count = zero)

Response: The variable is “Source of infection” with the following qualitative categories: Lung, Abdomen, Urinary tract, Skin-soft tissues, Catheter, Others. The p value comes from a Chi2 test for the overall variable covering all categories, and comparing them between survival and non-survival groups. 

- what were the results of cultures( blood, urine, etc):

Response: We have information about the source of infection but we do not have information about the specific microorganism responsible for the infection. We believe that given the small sample size the conclusions that can be drawn from this information are limited. Even so, if the reviewer considers this information indispensable, we could try to complete it.

- Ptx3 levels increase in other NON-SEPSIS condition as hypoxia, renal insult . DO you exclude those patients??

Response: Indeed, as explained in the section corresponding to methods, only patients who met criteria for septic shock were included in the study. No patient was included who did not have an infection as the origin of his symptoms. Therefore, patients with other symptoms were excluded.

- what is the initial diagnosis of your patients

Response: In all cases the initial diagnosis and the reason for ICU admission was septic shock.

- it is recommended to include figures of ROC

Response: We agree, we have performed a figure of ROC for all biomarkers including Lactate, as S Fig 1. It is showed below.

S Fig 1. ROC Curves for Severity Scores and biomarkers with respect to ‘in hospital mortality’.

Reviewer #2: In the present study, Perez-San Martin et al. present data showing the prognostic potential of PTX3 in a cohort of 75 septic shock patients. They state that adding PTX3 to severity scores increased the predictive capacity of the latter for mortality, and that a single determination of plasma PTX3 on ICU admission was a better prognostic marker compared with PCT and CRP.

The study is relatively limited in terms of number of patients involved, but it confirms the general messages of previous studies on the prognostic potential of PTX3 in sepsis.

Major points:

- The authors state that adding PTX3 to severity scores increased the predictive capacity of the latter for mortality. However, this increase is really limited and AUC reported in Tables 2 and 3 are almost identical. If there is any advantage, is this statistically significant?

Response: We agree that there is not a substantial increase between Tables 2 and 3 (adding PTX3), however we think that the fact that PTX3 increase the AUC of each severity score, deserve future attention, because for the other studied biomarkers (PCT and CRP), we did not find these findings. Further studies must deep in the clinical relevance of these findings, to elucidate whether an apparent low increase in AUC may be clinically relevant. We have added in the discussion a sentence about it. 

Now it reads: “The predictive accuracy in relation to severity scores was slightly increased by adding PTX3 to the logistic models. The clinical relevance of this slight increase deserve further attention in future studies, to elucidate whether systemic PTX3 levels would have prognostic value and may help to determine the prognosis of patients with septic shock, complementing disease severity classification systems and other biological markers”.

-The authors should better underline the novelties of this study compared to papers published in the last years on this marker in sepsis, which are collected in the review by Porte et al (Frontiers Immunol, 2019), in addition to ref 18. Among them, there are studies with larger sample size.

Response: We agree with the observation made by the reviewer. We have included a paragraph in this section that includes both the bibliographic citation and the articles to which the reviewer refers and whose results have been compared with our own.

-The authors discuss the difference with results reported in ref 12 and 15, pointing to the value of PTX3 levels at ICU admission found in this study, but not in the studies of ref 12 and 15, where the authors state that levels on day 5 and 7, respectively, correlated with severity, but PTX3 levels on day 1, did not. In this regard, it would be important to discuss criteria for ICU admission, and the time between sepsis diagnosis and PTX3 test, since these factors could explain the difference with previous studies.

Response: we agree with the comments made by the reviewer. It is possible that the factors cited influence the different results observed. For this reason we have included changes to the text in this regard and now it reads “These conflicting outcomes may due to the heterogeneity of different study subjects. Both studies were designed with patients with severe sepsis and septic shock according to the first criterion of sepsis based on the concept of systemic inflammatory response syndrome (SIRS). In addition, patients were randomized at enrollment. However, our clinical study was conducted on a homogeneous group of patients who met the updated definition of septic shock. On the other hand, the timing of mortality analysis was different (90-day mortality vs. in-hospital mortality). Furthermore, in both studies a sandwich ELISA developed in-house was used to determine PTX3 levels.”

-It would be important to add figures, e.g. for AUC and correlation analysis.

Response: We agree. We have performed a figure of ROC Curves for Severity Scores and biomarkers (including Lactate) with respect to ‘in hospital mortality’ (see please S Fig 1 and response to review 1).

-It would be interesting to perform further analysis based on the characteristics of patients. For instance, is there any difference in PTX3 levels/predictive capacity depending on the infection site?

Response: PTX3 mean and median levels, seems to be higher in patients with abdominal or lung infection than those with urinary tract infection. 

Pentraxin-3 (ng/ml) 

Source of infection

Mean

Median

N

SD

P25

P75

Lung

94,47

70,40

31

81,45

22,00

151,80

Abdomen

126,71

103,95

24

96,15

35,20

250,00

Urinary tract

41,32

14,30

9

78,91

4,60

29,70

Others

105,08

67,10

11

101,29

11,50

250,00

Total

99,97

63,80

75

91,05

24,20

187,00

Regarding predictive capacity depending on the infection site, as there is not events (zero in-hospital mortality) in urinary tract infection patients (see please table 1 and response to reviewer 1), it is not possible to perform a ROC curve for the specific category of urinary tract infection patients. On the other hand, a subgroup analysis would make the N even lower, which is another limitation. We agree with the reviewer that a subgroup analysis would enrich the analysis strategy. However, because of the aforementioned, we think it is preferable not to present these results in the manuscript.

Reviewer #3: The authors discuss the prognostic role of a single determination at the time of ICU admission of 3 biomarkers (PCT, CRP, PTX-3) in septic shock patients. They conclude that, after adjustment for confounding, PTX-3 has a prognostic value superior to the other two: the higher this biomarker, the higher the mortality.

I have several concerns.

1) The authors state that biomarkers are needed to stratify the patients according to the risk of death. Indeed, severity scores gather several clinical characteristic to convert them into a risk of mortality. PCT and CRP, being inflammatory markers, are certainly related to the prognosis, however their clinical use is usually different: early recognition of infection, follow-up of a certain treatment, antibiotic stewardship. Do you really think that such biomarkers could perform better than a severity score in prognostication? In Table 2 and 3 the authors present these findings, but they do not discuss them appropriatedly.

Response: We agree, due to the small sample size, the unicentric character, the existing bibliography, and the constructive arguments given by the reviewer, we have changed the conclusions and the discussion 

Now, the new text of conclusions in the abstract and discussión is:

“Abstract Conclusion: Our results support the prognostic value of a single determination of plasma PTX3 as a predictor of hospital mortality in septic shock patients”.

“Discussion conclusions: In summary, our results, support the available published studies suggesting the prognostic value of a single determination of plasma PTX3 as a predictor of hospital mortality in septic shock patients, defined according to the latest Sepsis-3 criteria. 

Further studies with larger sample size and multicentre character are needed in order to generalise these results, to elucidate the existence of subgroups of patients with specific characteristics in whom this biomarker could demonstrate higher accuracy in assessing the risk of mortality or increased severity, deepening finally in the knowledge the PTX3 predictive potential in comparison to other existing biomarkers such as Lactate, PCT or CRP or severity scores”.

In the discussion section, the new sentences referring to Table 2 and 3 results are:

“The predictive accuracy in relation to severity scores was slightly increased by adding PTX3 to the logistic models. The clinical relevance of this slight increase deserve further attention in future studies, to elucidate whether systemic PTX3 levels would have prognostic value and may help to determine the prognosis of patients with septic shock, complementing disease severity classification systems and other biological markers”.

2) The classical biomarker for sepsis prognostication is lactate, which is not discussed at all.

Response: We agree, we studied lactate with good results. As our main objective was to compare PTX3 with PCT and CRP we were undecided about including information on this classic biomarker. Now we have included this information. 

3) Premise: I am not a statistician. The ROC/AUC approach is ok, but I think that a survival analysis should be used rather than a logistic regression. The exposure-response trend with three groups does not add much to the analysis performed with two groups. Table 4 and 5 are redundant.

Response: Survival analysis is also a valid approach through a Cox regresión model. However, it estimates Hazard Ratios (HR) instead of Odds Ratios. We decided to use a logistic regression model because it estimates OR allowing for better comparisons with other studies from biomarkers. 

We agree that Table 4 is redundant with Table 5 in the sense that both tables show crude and adjusted ORs. However, as we have added Lactate results in Table 5 we think that combining both tables would produce a too large table, more difficult to read.

Additional remarks

- The biological and clinical role of the biomarkers should be clearly presented either in the Introduction or in the Discussion section without repetition

Response: we agree with the suggestion made by the reviewer. we have included the requested information in both sections.

- Try to discuss the discrepancies with Caironi's and Mauri's studies

Response: we agree with the suggestion made by the reviewer. We have modified this paragraph and now it reads “In contrast to our study, Mauri et al. reported that, although high PTX3 levels over the first 5 days from onset of sepsis were correlated with poorer outcomes, the initial PTX3 levels at ICU admission did not differ between survivors and non-survivors (12). Additionally, Caironi et al., in a large multicentre randomised controlled trial, found that plasma PTX3 concentrations were higher in non-survivors compared to survivors on day 1 and were correlated with severity. While PTX3 levels on day 7 showed a significant predictive value for 90-day mortality, PTX3 levels on day 1, after adjustment for all confounders, were not associated with 90-day mortality in patients with severe sepsis or septic shock”

- Lack of a non-septic control group is not a limitation of the study

Response: our idea was that in an ICU control group without sepsis, PTX-3 higher levels would be less associated with mortality, supporting the hypothesis of being a selective predictor in sepsis. Nevertheless, we agree that is not directly related with our main objective (focused on sepsis) and we have deleted this limitation.

- Conclusions, first sentence: PTX-3 is an actual, not potential prognostic factor. The sentence does not mean anything in this way.

Response: We agree we have changed conclusions in abstract and discussion section.

---

## [Editor Report · Decision Letter 1]

16 Nov 2020

PONE-D-20-28810R1

PROGNOSTIC VALUE OF PLASMA PENTRAXIN 3 LEVELS IN PATIENTS WITH SEPTIC SHOCK ADMITTED TO INTENSIVE CARE

PLOS ONE

Dear Dr. Suberviola,

Thank you for submitting your manuscript to PLOS ONE. After careful consideration, we feel that it has merit but does not fully meet PLOS ONE’s publication criteria as it currently stands. Therefore, we invite you to submit a revised version of the manuscript that addresses the following points:

- the language used in the revised sections of the manuscript needs to be improved. In particular: 

1. I suggest using an alternative phrase for describing/comparing the magnitude of AUC. Consider using "lower" instead of "smaller".

2. Please consider rephrasing the Conclusion sentence of the Abstract: instead of "Our results support the prognostic value", consider the following: "Our results suggest that the prognostic value of... may support..."

3.  Please rephrase the following: "But even they have limited abilities..."

4. Please, try rephrasing the following sentence: "Lactate was the biomarker with higher associations..."

5. "Our results also support the severity scores..." Please consider using "Our results also suggest that..."

6. Please consider rephrasing: "slight increase"

7. Please revise the following sentence: "These conflicting outcomes may due to the heterogeneity..."

I suggest proofreading/editing the manuscript from a proofreader with full professional proficiency in scientific English.

We look forward to receiving your revised manuscript.

Kind regards,

Aleksandar R. Zivkovic

Academic Editor

PLOS ONE

---

## [Author Response · Author response to Decision Letter 1]

27 Nov 2020

Attached to this email I am sending you a new manuscript that includes all the changes suggested by the reviewers. In addition, English has been reviewed by a professional translator specialized in scientific articles.

- The language used in the revised sections of the manuscript needs to be improved. In particular: 

1. I suggest using an alternative phrase for describing/comparing the magnitude of AUC. Consider using "lower" instead of "smaller".

We have made the suggested change.

2. Please consider rephrasing the Conclusion sentence of the Abstract: instead of "Our results support the prognostic value", consider the following: "Our results suggest that the prognostic value of... may support..."

We have made the suggested change

3. Please rephrase the following: "But even they have limited abilities..."

We have made the suggested change

4. Please, try rephrasing the following sentence: "Lactate was the biomarker with higher associations..."

We have made the suggested change

5. "Our results also support the severity scores..." Please consider using "Our results also suggest that..."

We have made the suggested change

6. Please consider rephrasing: "slight increase"

We have made the suggested change

7. Please revise the following sentence: "These conflicting outcomes may due to the heterogeneity..."

We have made the suggested change

We trust that the new manuscript meets the requested specifications and we apologize for any inconvenience that the inappropriateness of its format may have created.

Best regards.

Borja Suberviola

---

## [Editor Report · Decision Letter 2]

30 Nov 2020

PROGNOSTIC VALUE OF PLASMA PENTRAXIN 3 LEVELS IN PATIENTS WITH SEPTIC SHOCK ADMITTED TO INTENSIVE CARE

PONE-D-20-28810R2

Dear Dr. Suberviola,

We’re pleased to inform you that your manuscript has been judged scientifically suitable for publication and will be formally accepted for publication once it meets all outstanding technical requirements.

Kind regards,

Aleksandar R. Zivkovic

Academic Editor

PLOS ONE
---

## [Editor Report · Acceptance letter]

2 Dec 2020

PONE-D-20-28810R2 

Prognostic value of plasma pentraxin 3 levels in patients with septic shock admitted to intensive care 

Dear Dr. Suberviola:

I'm pleased to inform you that your manuscript has been deemed suitable for publication in PLOS ONE. Congratulations! Your manuscript is now with our production department. 

Kind regards, 

on behalf of

Dr. Aleksandar R. Zivkovic 

Academic Editor

PLOS ONE